# Towards Enterprise Sustainable Innovation Process: Through Boundary-Spanning Search and Capability Reconfiguration

**Ning Cao** [1,2]**, Jianjun Wang** [3,]*****, **Yulu Wang** [1] **and Li'e Yu** [4]

1    School of Business, Shanghai DianJi University, Shanghai 201306, China; caon@sdju.edu.cn (N.C.); wangyl@sdju.edu.cn (Y.W.)
2    School of Economics and Management, Tongji University, Shanghai 200091, China
3    College of Marine Culture and Law, Shanghai Ocean University, Shanghai 201306, China
4    School of Economics and Management, Huaibei Normal University, Huaibei 235000, China; yule@chnu.edu.cn
*    Correspondence: jianjunwang@shou.edu.cn

**Abstract:** In the open innovation environment, enterprise sustainable innovation is no longer the result of individual decision-making. Extensive contact with suppliers, customers, scientific research institutions, and other subjects for boundary-spanning knowledge search, absorption, and reconfiguration is considered a critical path to enterprise sustainable innovation. Studying the process of "how boundary-spanning search affects enterprise sustainable innovation" has become an urgent and valuable task. Therefore, based on an innovation search perspective, this study explored the path and mechanism of boundary-spanning search affecting enterprise sustainable innovation, revealed the intermediary effect of capability reconfiguration, and clarified the regulatory role of information technology (IT) governance. We also proposed an integrated model promoting enterprise sustainable innovation process. Using questionnaire data from manufacturing companies in China, this study empirically tested the proposed model hypothesis. The results demonstrated that all boundary-spanning searches (supply-side, demand-side, and cross-regional searches) positively and significantly impacted enterprise sustainable innovation. However, the effects of the search types varied. Capability reconfiguration played a partial intermediary role between boundary-spanning search and enterprise sustainable innovation. IT governance positively moderated the relationship between boundary-spanning search and enterprise capability reconfiguration, particularly between cross-regional search and enterprise capability reconfiguration. This study enriches our understanding of the sustainable innovation process and provides theoretical guidance for enterprises to improve their sustainable innovation performance by effectively using boundary-spanning search strategies.

**Keywords:** enterprise sustainable innovation process; boundary-spanning search; capability reconfiguration; investment decision

## 1. Introduction

The wave of cross-boundary cooperation and deep integration triggered by the new generation of information technology has penetrated into many fields of industry and enterprise operation [1,2]. In this complex and turbulent environment, if the competitive strategy, management system, and technical model that brought advantages to enterprises in the past are difficult to adapt to the needs of the new competitive environment, they often become the "core rigidity" restricting enterprise development, and the signal accidental innovation or short-time innovation cannot promote enterprises' sustainable growth [3]. Sustainable innovation characterized by openness and dynamics has become an important strategy to improve the survival rate, vitality, and high-quality development of enterprises, particularly for manufacturing enterprises that are climbing to the middle and high-end position of the "smile curve" of the industrial value chain [4,5].

Sustainable innovation is a complex systematic project that requires continuous factor investment and reconfiguration of various capabilities [6,7]. However, due to the complexity of innovation and the interdisciplinary nature of knowledge, most innovation resources are distributed outside the boundaries of the organization [8]. Therefore, cross-boundary search and integration of internal and external multidimensional innovation resources to reconstruct enterprise business processes and core competence systems have become the third way to improve the sustainable competitive advantage of enterprises in addition to internal R&D and external merger [9,10]. Therefore, studying the process of "how boundary-spanning search affects enterprise sustainable innovation" has become an urgent and valuable research topic.

Scholars have studied the concepts, influencing factors, and relationships related to sustainable innovation and boundary-spanning search. Clausen proposed that sustainable innovation is when enterprises constantly update and integrate technology, market, and management knowledge and other resources to obtain a competitive advantage [11]. Wassenhove emphasized that sustainable innovation is a complex system, which requires enterprise technologies, systems, processes, and supporting resources to break through traditional operating habits and create new competitive advantages through continuous collision and integration [12]. Some scholars have also studied the key factors affecting sustainable innovation and put forward means and mechanisms to improve enterprise sustainable innovation, such as increasing R&D inputs [13], listening to customers [14], and improving catch-up ability [15]. Other scholars have found that, in the era of the digital economy, alliances and cooperation among enterprises are common. Therefore, enterprise sustainable innovation is driven not only by endogenous variables but also by external factors. Enterprises can obtain sustainable competitiveness by optimizing the innovation environment [16], building multiple knowledge cooperation networks [17], and implementing boundary-spanning search strategies [18].

Of these factors, the boundary-spanning search for heterogeneous knowledge is considered a critical factor of enterprise sustainable innovation, and it has attracted the research interest of many scholars in recent years. Miceli stressed that boundary-spanning search can bring valuable innovative elements to an organization and improve its strategic agility and prosperity [19]. Sidhu noted that with the expansion of the search scope, enterprises are no longer satisfied with only using the upstream and downstream knowledge of the supply chain and local information; enterprises now tend to search for heterogeneous knowledge and capability modules across geospatial boundaries [20]. Therefore, studying the impact of search knowledge from different sources on enterprise sustainable innovation can provide valuable insights. Other scholars have argued that the impact of boundary-spanning search on sustainable innovation is not direct but occurs indirectly through other variables, such as reconstruction ability [21,22], opportunity identification [23], R&D orientation [24], and technology status [25]. Enterprise capability reconfiguration is regarded as an intermediary "bridge" for boundary-spanning search to affect sustainable innovation [26,27]. Therefore, revealing the mediating role of capability reconfiguration between boundary-spanning search and enterprise sustainable innovation is particularly significant. The study also found that leading manufacturing enterprises integrated into the digital economy actively, so information and technology (IT) not only provided real-time technical support for enterprises to embed innovation networks and acquire external knowledge [28] but also offered modern governance methods for enterprise internal and external relationship management [29,30]. Therefore, introducing IT governance as a situational variable to clarify whether IT governance plays a mediating role in boundary-spanning search and capability reconfiguration can enrich and improve the enterprise sustainable innovation model.

In summary, when tracking the development process of Chinese manufacturing enterprises in the past 30 years, we found that the common characteristics of leading manufacturing enterprises were sustainable progress, continuous transformation, and self-transcendence. They integrated into the digital economy actively, connected domestic and foreign enterprises widely, and carried out boundary-spanning search and capability recon-

figuration continuously, so as to promote the sustainable process of innovation. However, the existing literature on how manufacturing enterprises affect the process of enterprise sustainable innovation through boundary-spanning search is still relatively weak. Therefore, this study takes manufacturing enterprises as the research objects and explores the path and mechanism of the effects of boundary-spanning search on sustainable innovation in enterprises from the perspective of organizational search and capability reconstruction to reveal the intermediary role of capability reconfiguration between boundary-spanning search and enterprise sustainable innovation and clarify the moderating role of IT governance. This study can enrich the innovation search theory, broaden the research vision of the driving factors of sustainable innovation in manufacturing enterprises, and open the "black box" of how the external knowledge is internalized and applied to the new knowledge creation. The research also provides the theoretical guidance for enterprises to break through knowledge limitations and capacity constraints in order to effectively improve sustainable innovation performance through boundary-spanning search strategies.

## 2. Literature Review and Research Hypothesis

### 2.1. Boundary-Spanning Search

Enterprise sustainable innovation requires interdisciplinary knowledge. Boundary-spanning search is a crucial method for enterprises to obtain diversified resources and knowledge [31]. Scholars have studied the connotation, boundary, and classification dimensions of boundary-spanning search and their relationship with innovation performance. Rosenkopf explained that boundary-spanning search is related to boundary management. Through boundary-spanning search, enterprises can obtain heterogeneous knowledge and information outside their professional field in a complex and dynamic environment [32]. Yang demonstrated that boundary-spanning search, as the driving factor of enterprise sustainable innovation, helps enterprises overcome the lack of innovation resources and allows them to perceive the capability gap, find business opportunities, and enhance sustainable competitiveness [33]. Some scholars take the "search scope" as the standard, dividing boundary-spanning searches into broad and narrow searches [34]. In contrast, other scholars take "search content" as the standard, dividing boundary-spanning searches into market knowledge and technical knowledge searches [35,36], or the "search method" as the standard, dividing boundary-spanning searches into interactive and noninteractive searches [37]. Among the classification approaches, the Sidhu division method is the most widely used in practice because it identifies the subject of the knowledge search. He proposed that enterprises look for knowledge not only from the supply and demand sides of their region and industry but also from other industries, regions, and countries [20]. Referring to the Sidhu classification, this study divided boundary-spanning searches into supply-side, demand-side, and cross-regional searches. Here, the supply-side search primarily obtains knowledge from suppliers and scientific research institutes, while the demand-side search predominately obtains knowledge from customers, dealers, and competitors. The cross-regional search primarily obtains specialized technology and scarce knowledge from outside the region or industry.

### 2.2. Enterprise Sustainable Innovation Performance

Research on enterprise sustainable innovation primarily focuses on its causes, definition, and evaluation. Suarez proposed that sustainable innovation exhibits the characteristics of continuous accumulation and systematic change. It is a process where enterprises update their core competence system and transform it into a future competitive advantage [38]. Lianto argued that enterprises with successful innovation experiences also succeed with subsequent innovation processes, which is defined as enterprise sustainable innovation [39]. Nam studied the innovation of small- and medium-sized enterprises in Vietnam and found that sustainable innovation is a process in which new ideas, technologies, products, and markets are constantly generated [40]. The evaluation of sustainable innovation is typically conducted along two primary lines: process and results. On the

one hand, enterprise sustainable innovation performance is reflected by the R&D inputs, organization management level, transformation ability of technological achievements, expansion ability of external relations [41], etc. On the other hand, enterprise sustainable innovation performance can also be reflected by the growth of a product, technology, market, management, and service and their lagging effects [42]. This study comprehensively applies these indicators to evaluate the performance of enterprise sustainable innovation.

### 2.3. Capability Reconfiguration

With the acceleration of industrial division and integration, enterprises are facing a series of pressures, such as accelerated technological change, intensified market volatility, and frequent random incidents. Enterprises must recognize and regenerate new knowledge systems to support their innovation sustainability [26]. Utoyo defined capability reconfiguration as brokerage and restart. It is a process in which enterprises take the initiative to change the original knowledge system, reshape internal and external relationships, and cultivate a more valuable knowledge system [43]. Enterprise capability reconfiguration runs through the entire innovation process of factor combination, optimization, and effective operation. Thus, enterprise capability reconfiguration allows enterprises to review their current knowledge system; retain existing competitive advantages; continuously absorb and utilize new knowledge elements; and realize capability renewal, replacement, and redeployment [44]. Relevant studies have found that capability reconfiguration is critical to enterprise innovation.

### 2.4. IT Governance

IT provides real-time technical support for enterprises to embed multiple knowledge networks for interconnection and information exchange. It has also become a modern governance mechanism to coordinate the internal and external relationships of enterprises [28]. Chi emphasized that IT governance is a series of structured arrangements based on IT applications that aims to securely retrieve, store, and transfer key information [29]. With the popularity of alliances and cooperation among enterprises, IT governance mechanisms have gradually been enriched, including a series of structural and institutional arrangements such as information channel diversification, information storage standardization and networking, information diffusion platform creation, and information application modularization [29]. Ko argued that IT governance plays a role in value creation, risk control, and strategic coordination, It supports cross-border cooperation among organizations, reduces the information flow risk among enterprises, and promotes small- and medium-sized enterprise innovation [45]. Therefore, many enterprises have designed and applied IT governance to enterprise innovation activities and relationship arrangements.

### 2.5. Boundary-Spanning Search Effect on Enterprise Sustainable Innovation

According to organizational search theory, boundary-spanning search increases the knowledge stock of enterprises and reduces the dual constraints of resource shortage and weak ability. The acquired knowledge can help enterprises find opportunities, solve problems, learn and cultivate new skills, and promote adaptive growth [46]. The complementary technical knowledge acquired by enterprises from suppliers and scientific research institutions can improve the success rate of enterprise technological changes and shorten the innovation cycle [47,48]. Supply-side search is also conducive to enterprises overcoming the bottleneck of technological development, changing the inertial development track, and realizing reorganization innovation [49–51]. Zimmermann proposed that enterprises must listen to their customers on the demand side, track the latest changes in the market, and ensure that innovations meet the market demand [52]. Thus, demand-side search can deepen enterprise understanding of customer consumption habits and competitor business models and enhance enterprise ability to adapt to the external environment and provide appropriate products or services [53]. To break organizational inertia and capability rigidity, Meulman found that the search across enterprise boundaries should not be limited to local

partners but should also include the expertise of global peer enterprises, competitors, and other remote partners [54]. This is because the "close neighbor dependence" on local suppliers and customers can lead to enterprise "strategic shortsightedness" [55]. Therefore, promptly tracking cross-regional knowledge can provide scarce and heterogeneous resources for enterprises readjusting business models or developing new products, allowing them to reach middle or high-end status in the value chain [56]. Thus, this study puts forward the following hypotheses:

**Hypothesis 1a.** *Supply-side search positively influences enterprise sustainable innovation.*

**Hypothesis 1b.** *Demand-side search positively influences enterprise sustainable innovation.*

**Hypothesis 1c.** *Cross-regional search positively influences enterprise sustainable innovation.*

### 2.6. Mediating Effect of Capability Reconfiguration

According to the knowledge-based view, sustainable innovation is when enterprises continuously accumulate innovation elements and update existing competitive advantages to benefit future innovation [38]. In an open and dynamic environment, the internal and external innovation environments are complex, changeable, and uncertain. Only by constantly updating the core competitiveness structure can enterprises support their sustainable innovation activities [57]. Dutt discussed the antecedents and consequences of capability reconfiguration. He proposed that the knowledge obtained by enterprises cannot be simply added together. Enterprises need to understand the characteristics of various knowledge types and code and reorganize them to absorb and apply them to their innovation system and maximize the value of the new knowledge elements [58]. Konlechner argued that boundary-spanning search triggers organizational capability replication, updating, replacement, and upgrading. Therefore, enterprise capability reconfiguration is a bridge connecting boundary-spanning search and enterprise sustainable innovation [26]. From the classification perspective, the knowledge obtained from supply-side, demand-side, and cross-regional searches by enterprises provides a driving force for enterprises to break their inherent cognitive limitations and expand, integrate, and reposition their capabilities. Further, capability reconfiguration promotes the innovation of new technology paradigms and business models [59]. Therefore, capability reconfiguration plays an intermediary role in boundary-spanning search affecting enterprise sustainable innovation. Thus, this study puts forward the following hypotheses:

**Hypothesis 2a.** *Capability reconfiguration plays a mediating role between supply-side search and enterprise sustainable innovation.*

**Hypothesis 2b.** *Capability reconfiguration plays a mediating role between demand-side search and enterprise sustainable innovation.*

**Hypothesis 2c.** *Capability reconfiguration plays a mediating role between cross-regional spatial search and enterprise sustainable innovation.*

### 2.7. Moderating Effect of IT Governance

The popularization and application of IT have greatly impacted the innovation catch-up of enterprises [60]. Particularly in the open innovation environment, organizations with IT arrangements and an IT governance framework can establish virtual cooperative relationships, communicate and share information 24 h a day, and obtain a wide range of global knowledge [61]. When an enterprise IT governance level was high, the enterprise had established a mutually supportive communication mechanism and sharing platform. The information exchange between different subjects exhibited unified coding and sharing rules, and the speed of information processing was accelerated. This, it is easier for enterprises to obtain knowledge and information from the supply side, demand side, and

across regions; the search costs are reduced, and the reconstructing efficiency is improved. When the level of IT governance was low, the cost and risk of obtaining external information were high, weakening the boundary-spanning search, particularly the cross-regional search, and reducing its impact on enterprise capability reconfiguration. Therefore, this study proposes the following hypotheses:

**Hypothesis 3a.** *IT governance positively moderates the relationship between supply-side search and enterprise capability reconfiguration.*

**Hypothesis 3b.** *IT governance positively moderates the relationship between demand-side search and enterprise capability reconfiguration.*

**Hypothesis 3c.** *IT governance positively moderates the relationship between cross-regional search and enterprise capability reconfiguration.*

*2.8. Moderated Mediating Effect of IT Governance*

Based on the above hypotheses, this study proposed that different IT governance levels enhance or weaken the impact of boundary-spanning search on enterprise capability reconfiguration and adjust the intermediary effect of capability reconfiguration between boundary-spanning search and enterprise sustainable innovation. When the IT governance level of the enterprise is high, the impact of boundary-spanning search on enterprise capability reconfiguration is enhanced; i.e., with the help of IT governance, enterprises can more easily use the supply-side, demand-side, and cross-regional search strategies to obtain valuable external knowledge to promote the renewal, iteration, and reconfiguration of the enterprise capability modules more efficiently. When enterprise capability is updated and reconstructed, it plays a more powerful role in promoting enterprise sustainable innovation. Therefore, this study puts forward the following hypothesis:

**Hypothesis 4.** *IT governance positively moderates the mediating effect of enterprise capability reconfiguration on the relationship between boundary-spanning search and enterprise sustainable innovation.*

After summarizing the above research assumptions, we found that manufacturing enterprises integrated into the digital economy actively, contacted domestic and foreign enterprises widely, carried out boundary-spanning search and capability reconfiguration, and promoted enterprise sustainable innovation and self-transcendence constantly. Therefore, based on the definition of key variables and their internal relationship, we built a comprehensive theoretical model among boundary-spanning search, capability reconfiguration, IT governance, and sustainable innovation of manufacturing enterprises, as shown in Figure 1. According to this model, we analyze the direct impact of supply-side, demand-side, and cross-regional searches on enterprise sustainable innovation; reveal the mediating effect of capability reconfiguration between boundary-spanning search and enterprise sustainable innovation; and evaluate the possible moderating effect of IT governance between boundary-spanning search and capability reconfiguration.

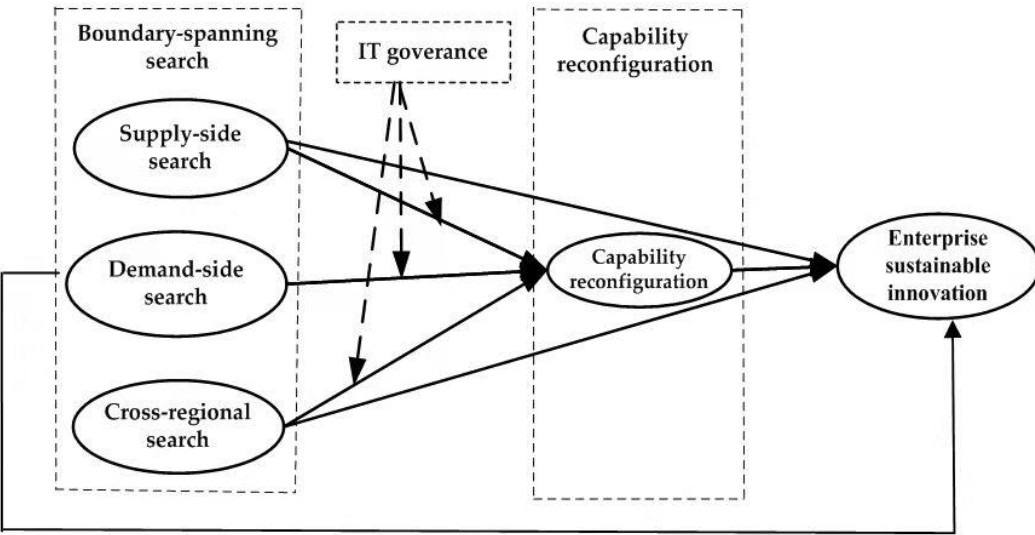

**Figure 1.** The theoretical model.

## 3. Research Methods

In this study, SPSS 22.0 and AMOS 22.0 statistical software were used to analyze the sample distribution, the reliability and validity of variables, and the statistics and correlation of variables. Then, multiple linear regression analysis and structural equation model analysis were used to analyze the mechanism by which boundary-spanning search, capability reconfiguration, and IT governance act on enterprise sustainable innovation.

### 3.1. Sample and Data Collection

After tracking the development process of leading manufacturing enterprises in emerging economies, we found that the common characteristics of these enterprises were sustainable innovation and continuous self-transcendence. This study focused on the impact of boundary-spanning search on the sustainable innovation of manufacturing enterprises in emerging economies. Therefore, questionnaires were distributed to Chinese manufacturing enterprises with active innovation intentions, primarily including manufacturing enterprises in the electronic information, biomedicine, automotive, aerospace, and high-end equipment manufacturing industries. To ensure the quality of the survey, the questionnaires were primarily distributed to middle and senior managers who are familiar with the overall situation of the company, and the respondents were required to have a sufficient understanding of enterprise innovation activities.

The questionnaire was predominantly distributed through EMBA students and alumni of Shanghai Jiaotong University and Tongji University, and it was primarily collected through on-site filling, online filling, and e-mail. From October 2020 to March 2021, 580 questionnaires were distributed, and 269 valid questionnaires were recovered, with an effective recovery rate of 57.60%. Among these, 375 questionnaires were distributed on-site, of which 314 questionnaires were recovered and 159 questionnaires were deemed valid (approximately 50.64% rate of effective recovery); 78 questionnaires were distributed via e-mail, of which 62 questionnaires were recovered and 38 questionnaires were considered as valid (about 61.29% rate of effective recovery) [62]; and 127 questionnaires were distributed via network, of which 108 questionnaires were recovered and 72 questionnaires were considered as valid (about 66.67% rate of effective recovery). The details of sample distribution and recovery are shown in Table 1.

**Table 1.** Sample distribution and recovery.

| Distribution Method | Issued Quantity | Recovery Quantity | Effective Quantity | Effective Recovery Rate |
|---|---|---|---|---|
| On-site distribution | 375 | 314 | 159 | 50.64% |
| E-mail distribution | 78 | 62 | 38 | 61.29% |
| Network distribution | 127 | 108 | 72 | 66.67% |
| Total | 580 | 467 | 269 | 57.60% |

Note: Effective recovery rate = effective quantity/recovery quantity.

According to the responses regarding the industries to which the enterprises belonged, the electronic information industry accounted for 27.88% of the total, the biomedical industry accounted for 26.76%, the automotive and aerospace industry accounted for 20.82%, the equipment manufacturing industry accounted for 14.50%, and other industries accounted for 10.04%. The specific conditions of the enterprise sample in this study are presented in Table 2.

**Table 2.** Distribution characteristics of the enterprise sample.

| Sample Characteristics | Category | Number | Percentage |
|---|---|---|---|
| Enterprise age | No more than 3 years | 8 | 2.97% |
| | 3–5 years | 37 | 13.75% |
| | 6–10 years | 95 | 35.32% |
| | 11–15 years | 83 | 30.86% |
| | More than 15 years | 46 | 17.10% |
| Enterprise scale | Fewer than 100 employees | 15 | 5.58% |
| | 101–500 employees | 29 | 10.78% |
| | 501–1000 employees | 60 | 22.30% |
| | 1001–2000 employees | 98 | 36.43% |
| | More than 2000 employees | 67 | 24.91% |
| Industries | Electronic information | 75 | 27.88% |
| | Biomedical | 72 | 26.76% |
| | Automotive and aerospace | 56 | 20.82% |
| | Equipment manufacturing | 39 | 14.50% |
| | Other industries | 27 | 10.04% |

*3.2. Variable Measurement*

All variables in this study were measured using a maturity scale widely applied by many scholars and improved through on-site interviews. All measurement indicators were scored by the Likert five-point scoring method (from 1 = strongly disagree to 5 = strongly agree).

(1) Measurement of the independent variable boundary-spanning search. We used the study by Sidhu [20] to measure the boundary-spanning search. The scale included 12 items, and its primary purpose was to reflect the current situation of enterprise supply-side, demand-side, and cross-regional searches.

(2) Measurement of the dependent variable enterprise sustainable innovation performance. Since enterprise sustainable innovation was a dynamic variable, we referred to the scales of Deschryvere [41] and Triguero [42] and selected five items to reflect the degree of enterprise sustainable innovation. We mainly adopted indicators that can reflect the growth rate of the enterprise's innovation input and output compared with the previous year, such as the growth rate of R&D personnel and the growth rate of new product sales revenue; we also adopted indicators that can reflect the relative growth rate of the enterprise's innovation investment and innovation achievements compared with the industry competitors, such as R&D investment, number of patent applications, and the growth rate of new market share.

(3) Measurement of the mediating variable capability reconfiguration. We primarily referred to the scales of Konlechner [26] and Subramanian [44] to measure enterprise capability reconfiguration. There were four items in the scale, which required respondents to objectively evaluate the enterprise capability renewal, replacement, redeployment, and upgrading [44].

(4) Measurement of the moderating variable IT governance. We mainly adopted the scales of Chi [29] and John [63] to measure the IT governance of enterprises. The scale includes four items, namely the diversification of information channels, standardization of information storage, platform creation of information diffusion, and modularization of information application.

Because larger and older enterprises may have accumulated greater absolute amounts of innovation resources, scale and age may affect the innovation achievements of the enterprise [64]. Therefore, this study took the scale and age of the enterprises as control variables.

*3.3. Reliability and Validity Analyses*

SPSS 22.0 and AMOS 22.0 statistical software were used to test the reliability and validity of the variables. By observing the data of analysis, it can be seen that the Cronbach's α values between each variable exceed the critical value of 0.70, indicating that the questionnaire exhibits sufficient reliability. The scales used in this study referred to the mature scale that has been used, and they were improved through on-site interviews. Thus, the content validity of the scale was also sufficient. Confirmatory factor analysis was performed on the scale; the standardized factor load of each variable was greater than 0.50, the average variance extraction (AVE) value was greater than 0.50, and the combined reliability value was greater than 0.60, indicating that the scale had convergence validity. Through confirmatory factor analysis, this study also found that the square root of each variable AVE value was greater than the Pearson correlation coefficient of its row and column, indicating that the scale had discriminate validity [65]. Compared with other competition models, this study found that an integration model with six factors exhibited the optimal fitting effect, as shown in Table 3 (the fitting indexes are as follows: $\chi^2/$Df = 1.534, GFI = 0.923, CFI = 0.907, TLI = 0.950, and RMSEA = 0.047), demonstrating that the six variables in the model exhibit sufficient discriminate validity and belong to different constructs.

**Table 3.** Fitting indexes of six-factor integration model.

|  | $\chi^2/$**Df** | **GFI** | **CFI** | **TLI** | **RMSEA** |
|---|---|---|---|---|---|
| Fitting index | 1.534 | 0.923 | 0.907 | 0.950 | 0.047 |
| Reference value | 1–3 | >0.90 | >0.90 | >0.90 | <0.08 |

## 4. Results

*4.1. Descriptive Statistical and Correlation Analysis*

SPSS 22.0 statistical software was used for descriptive and correlation analyses of the questionnaire data. The analysis results of the variable mean, standard deviation, and correlation coefficient are displayed in Table 4.

There was a positive correlation between supply-side search and enterprise sustainable innovation (correlation coefficient r = 0.492, $p < 0.01$). There was also a significant correlation between demand-side search and enterprise sustainable innovation (correlation coefficient r = 0.464, $p < 0.01$). In addition, the correlation coefficient between cross-regional search and enterprise sustainable innovation was significant and positive (correlation coefficient r = 0.418, $p < 0.01$), and the correlation coefficient between capability reconfiguration and enterprise sustainable innovation was significant and positive (correlation coefficient r = 0.535, $p < 0.01$). The correlation coefficient between IT governance and enterprise sustainable innovation was also significant and positive (correlation coefficient r = 0.350, $p < 0.01$). Thus, the correlation coefficients among the variables were all less than 0.70.

These analysis results provided preliminary support for the research hypotheses proposed in this study.

**Table 4.** Descriptive statistics and correlation analysis results ($n = 269$).

| Variable | 1 | 2 | 3 | 4 | 5 | 6 | 7 | 8 |
|---|---|---|---|---|---|---|---|---|
| 1. Enterprise age | — | | | | | | | |
| 2. Enterprise scale | 0.365 * | — | | | | | | |
| 3. Supply-side search | 0.130 * | 0.031 | 0.746 | | | | | |
| 4. Demand-side search | 0.112 | 0.022 | 0.572 ** | 0.732 | | | | |
| 5. Cross-regional search | 0.167 | 0.098 | 0.560 ** | 0.529 ** | 0.708 | | | |
| 6. Capability reconfiguration | 0.022 | 0.062 | 0.487 ** | 0.461 ** | 0.413 ** | 0.749 | | |
| 7. IT governance | 0.153 * | −0.109 | 0.290 ** | 0.247 ** | 0.267 ** | 0.423 ** | 0.781 | |
| 8. Sustainable innovation | 0.175 * | −0.139 | 0.492 ** | 0.464 ** | 0.418 ** | 0.535 ** | 0.350 ** | 0.778 |
| Mean | 2.870 | 3.060 | 3.810 | 3.880 | 3.800 | 3.780 | 3.950 | 3.952 |
| SD | 1.138 | 0.791 | 0.698 | 0.851 | 0.967 | 1.059 | 1.028 | 0.983 |

Note: Diagonal in the table refers to root square of AVE; * $p < 0.05$ and ** $p < 0.01$ (bilateral test).

### 4.2. Hypotheses Testing

The theoretical model and relevant hypotheses are verified, and the test results are displayed in Table 5.

**Table 5.** Model hierarchical regression results.

| Variable | Capability Reconfiguration | | | | Enterprise Sustainable Innovation | | | |
|---|---|---|---|---|---|---|---|---|
| | Model1 | Model 2 | Model 3 | Model 4 | Model 5 | Model 6 | Model 7 | Model 8 |
| Control variables | | | | | | | | |
| Enterprise age | 0.24 | 0.22 | 0.20 | 0.19 | 0.24 | 0.23 | 0.19 | 0.16 |
| Enterprise scale | 0.07 | 0.05 | 0.03 | 0.02 | 0.09 | 0.06 | 0.04 | 0.03 |
| Independent variables | | | | | | | | |
| Supply-side search | | 0.24 *** | 0.18 *** | 0.21 *** | | 0.38 *** | | 0.13 ** |
| Demand-side search | | 0.19 ** | 0.16 ** | 0.20 ** | | 0.33 ** | | 0.12 * |
| Cross-regional search | | 0.12 ** | 0.10 ** | 0.16 ** | | 0.15 ** | | 0.09 * |
| Mediator variable | | | | | | | | |
| Capability reconfiguration | | | | | | | 0.41 ** | 0.23 ** |
| Moderator variable | | | | | | | | |
| IT governance (ITG) | | | 0.14 *** | 0.15 ** | | | | |
| Interactions | | | | | | | | |
| ITG*supply-side search | | | | 0.09 ** | | | | |
| ITG*demand-side search | | | | 0.08 ** | | | | |
| ITG*cross-regional search | | | | 0.16 ** | | | | |
| $R^2$ | 0.06 | 0.35 | 0.43 | 0.47 | 0.25 | 0.28 | 0.33 | 0.38 |
| Adjusted $R^2$ | 0.05 | 0.34 | 0.42 | 0.45 | 0.24 | 0.26 | 0.31 | 0.36 |

Note: * $p < 0.05$, ** $p < 0.01$, and *** $p < 0.001$.

#### 4.2.1. Main Effect Test

Taking enterprise sustainable innovation as the dependent variable, this study verified the regression results of the control and independent variables on the dependent variable to obtain Models 5 and 6, respectively, as presented in Table 5. It can be seen from Model 5 that the control variables (enterprise age and enterprise scale) have no significant impact on enterprise sustainable innovation. Further, it can be seen from Model 6 that supply-side search ($\beta = 0.38$, $p < 0.001$), demand-side search ($\beta = 0.33$, $p < 0.01$), and cross-regional

search (β = 0.15, *p* < 0.01) all exhibited a positive and significant impact on enterprise sustainable innovation. Thus, Hypotheses 1a, 1b, and 1c were verified.

### 4.2.2. Mediating Effect Test

Based on the mediating effect analysis steps proposed by Baron [66], this study tests the mediating effect of capability reconfiguration between boundary-spanning search and enterprise sustainable innovation. The research results are presented in Table 5. According to the analysis data of Model 7, capability reconfiguration exhibited a positive effect on enterprise sustainable innovation (β = 0.41, *p* < 0.01). Comparing Model 8 with Model 6, it was found that the direct impact of supply-side search on enterprise sustainable innovation decreased from 0.38 to 0.13, but it was still significant (*p* < 0.01), indicating that capability reconfiguration played a partial intermediary role between supply-side search and enterprise sustainable innovation. The direct impact of demand-side search on enterprise sustainable innovation decreased from 0.33 to 0.12, but it remained significant (*p* < 0.05), indicating that capability reconfiguration played a partial intermediary role between demand-side search and enterprise sustainable innovation. The impact of cross-regional search on enterprise sustainable innovation decreased from 0.15 to 0.09 but was still significant (*p* < 0.05), indicating that capability reconfiguration also played a partial intermediary role between cross-regional search and enterprise sustainable innovation. Thus, Hypotheses 2a, 2b, and 2c were verified.

In order to further verify the mediation effect of capability reconfiguration, the Sobel test method was introduced in this study. The test results illustrated that capability reconfiguration exhibited a significant intermediary effect between supply-side search and enterprise sustainable innovation (Z = 3.02, *p* < 0.01). Additionally, capability reconfiguration demonstrated a significant intermediary effect between demand-side search and enterprise sustainable innovation (Z = 2.89, *p* < 0.01), and it exhibited a significant intermediary effect between cross-regional search and enterprise sustainable innovation (Z = 2.75, *p* < 0.01). Therefore, Hypotheses 2a, 2b, and 2c were further supported.

### 4.2.3. Moderating Effect Test

Taking enterprise capability reconfiguration as the dependent variable, a regression model was established. The control variable, independent variable, moderating variables, and the interactions between the moderating and independent variables were added in turn to the regression model to obtain Models 1 to 4, respectively. Model 4 demonstrated that the interaction between IT governance and supply-side search positively and significantly impacted enterprise capability reconfiguration (β = 0.09, *p* < 0.01), indicating that the higher the level of IT governance, the stronger the role of supply-side search in promoting enterprise capability reconfiguration. Therefore, Hypothesis 3a was supported. The interaction between IT governance and demand-side search also positively and significantly impacted enterprise capability reconfiguration (β = 0.08, *p* < 0.01), indicating that the higher the level of IT governance, the stronger the role of demand-side search in promoting enterprise capability reconfiguration. Thus, Hypothesis 3b was also supported. The interaction between IT governance and cross-regional search also exhibited a positive and significant impact on enterprise capability reconfiguration (β = 0.16, *p* < 0.01), indicating that IT governance promoted the positive impact of cross-regional search on enterprise capability reconfiguration and verifying Hypothesis 3c. This study further found that IT governance demonstrated a greater moderating effect on cross-regional search than supply-side and demand-side searches. Therefore, Hypotheses 3a, 3b, and 3c were verified.

### 4.2.4. Moderated Mediating Effect Test

This study used the bootstrap test method [67] to verify the moderated mediating effect. The test results are displayed in Table 6. When the enterprise IT governance level was low (expressed by the mean minus one standard deviation), the mediating effect of enterprise capability reconfiguration was not significant, and the 95% confidence interval

ranged from −0.017 to 0.072 (including 0). When the enterprise IT governance level was high (expressed by the mean plus one standard deviation), the mediating effect value of enterprise capability reconfiguration was 0.135, and the 95% confidence interval ranged from 0.033 to 0.254 (excluding 0), indicating that there was a moderated mediating effect. IT governance positively moderates the mediating effect of capability reconfiguration on the relationship between boundary-spanning search and enterprise sustainable innovation. Thus, Hypothesis 4 was verified.

**Table 6.** Moderated mediating effect test with bootstrap analysis.

| Mediator Variable | IT Governance | Indirect Effect | SE | 95% CI | |
|---|---|---|---|---|---|
| | | | | LLCI | ULCI |
| | 2.922 (M − 1SD) | 0.023 | 0.024 | −0.017 | 0.072 |
| Capability reconfiguration | 3.950 (M) | 0.119 ** | 0.034 | 0.027 | 0.225 |
| | 4.978 (M + 1SD) | 0.135 ** | 0.041 | 0.033 | 0.254 |

Note: ** $p < 0.01$; bootstrap = 5000.

## 5. Discussion

### 5.1. Results Discussion

When tracking the development process of Chinese manufacturing enterprises in the past 30 years, we found that the common characteristics of leading manufacturing enterprises were sustainable innovation, continuous transformation, and self-transcendence. They integrated into the digital economy actively, connected domestic and foreign enterprises widely, and carried out boundary-spanning search and capability reconfiguration continuously, so as to promote the sustainable process of innovation. Therefore, based on innovation search theory, this study explored the mechanisms of boundary-spanning search affecting enterprise sustainable innovation, verified the mediating effect of capability reconfiguration and the moderating effect of IT governance, and obtained the following research conclusions:

(1) Boundary-spanning search (including supply-side, demand-side, and cross-regional searches) positively and significantly impacted enterprise sustainable innovation; however, the effects of the three search types were different. This may be because the cost and implementation difficulty of supply-side and demand-side searches are lower than those of cross-regional searches; thus, supply-side and demand-side searches play a greater role in promoting enterprise sustainable innovation. Cross-regional searches can bring more valuable market information and complementary technical knowledge to enterprises. However, they are more difficult to implement, and the cost of communication and coordination is higher. Therefore, in the earlier stage, enterprises tend to adopt more local search strategies. By forming partnerships with suppliers, scientific research institutes, and customers near the primary operating location, the enterprise can search for advanced technologies. Taking advantage of the trust accumulated over time through interactions, enterprises can find favorable business opportunities and promote sustainable innovation. When the enterprise grows to a certain size and the global market becomes the primary competitive landscape, the enterprise can transfer the search center to global value networks and cross-regional partners. This research conclusion is consistent with the view that "local search serves as a springboard for international search of late developing enterprises" put forward by Huggins [56].

(2) Capability reconfiguration is the intermediary bridge between boundary-spanning search and enterprise sustainable innovation. Through the mediating test, we found that the impact of boundary-spanning search on enterprise sustainable innovation was partly realized through capability reconfiguration. Capability reconfiguration is critical for enterprises to replace, repair, and redeploy their capability system and innovative elements. Particularly in recent years, the innovation resources required for enterprises have become

more modular, and it is common to participate in modular cooperative innovation for enterprises. Hence, enterprises should pay greater attention to the reconfiguration of core competencies, which is the key to promoting enterprise sustainable innovation. This research conclusion is consistent with the statement by Utoyo that "sustainable innovation comes from the reconstruction, iteration and renewal of enterprise core competence" [16].

(3) IT governance exhibits a positive moderating effect on the relationship between boundary-spanning search and capability reconfiguration, particularly the moderating effect on cross-regional search and capability reconfiguration. This result demonstrates that with an improvement in the enterprise IT governance level, boundary-spanning search enhances enterprise capability reconfiguration. When enterprises seek heterogeneous knowledge through cross-regional search, effective IT governance is needed to encode, transmit, and spread novel knowledge to promote the iteration and sublimation of the enterprise core competitiveness. This finding is consistent with the view put forward by Chi that "IT governance structure promotes the core value creation of organization" [29].

### 5.2. Theoretical Contributions

This study has the following theoretical contributions: (1) When enterprise sustainable innovation occurs in an open and interactive environment, from the perspective of multidimensional knowledge search, supply-side, demand-side, and cross-regional searches can improve enterprise sustainable innovation performance. These results enrich the classification research of organizational boundary-spanning search and expand the driving factors of enterprise sustainable innovation. (2) This study verified the intermediary role of enterprise capability reconfiguration and determined the decisive role of capability reconfiguration in improving enterprise sustainable innovation. It also revealed how external diversified knowledge affects the "black box" of the enterprise sustainable innovation intermediary mechanism through capability reconfiguration. (3) This study revealed the positive moderating effect of IT governance on the relationship between boundary-spanning search and capability reconfiguration, further enriching the research on enterprise IT governance and providing a situational boundary for enterprises to adopt a boundary-spanning search strategy and improve the level of knowledge reconfiguration in the information age.

### 5.3. Managerial Enlightenment

The research conclusions lead to the following recommendations for enhanced enterprise sustainable innovation management:

(1) Enterprises should pay attention to their application of boundary-spanning search strategy. In an open innovation environment, boundary-spanning search is strategically significant for enterprises. It increases the stock, diversification, and novelty of enterprise knowledge; promotes the reconstruction of high-level enterprise ability; and fundamentally supports sustainable innovation. Manufacturing enterprises can formulate policies and strategies for supply-side, demand-side, and cross-regional searches according to the internal and external environmental conditions and internal knowledge resource base. For example, enterprises can form alliances and cooperation networks among supply chains, industrial chains, and international enterprises; establish knowledge sharing and information exchange platforms among organizations; and improve knowledge coding and storage mechanisms. With a change in the external environment, enterprises should promptly update the resource search type, save the retrieved knowledge, and form a standardized knowledge base to provide sufficient supporting resources for enterprise capability reconfiguration.

(2) Enterprises should also pay attention to the dynamic reconfiguration of their capability. The scarcity of innovation resources forces enterprise managers to dynamically balance and reconstruct existing and new resources and internal and external resources. However, relevant studies demonstrate that most Chinese enterprises have an insufficient reconfiguration capability. Due to organizational inertia, many enterprises remain in the low-level stage of "taking and using". Therefore, enterprises must cultivate dynamic

reconfiguration ability. Enterprise managers can change the organizational structure, incentive mechanism, corporate culture, participant skills, and strategic agility to promote the renewal, reconstruction, and upgrade of enterprise capability, which is vital for enterprises to obtain a long-term competitive advantage.

(3) The IT governance level of enterprises should be improved. Enterprises should establish diversified information channels and standardized information coding rules and actively apply IT governance to internal and external relationship management and knowledge network integration. In addition, enterprises can establish a platform for information dissemination, allowing IT governance to serve as the alliance and cooperation strategy among enterprises; provide services for external information retrieval and internal information storage and flow; and play a critical role in value creation, risk control, and strategic synergy.

*5.4. Limitations and Future Research*

There are some limitations of this study. First, boundary-spanning searches, including supply-side, demand-side, and cross-regional searches, exert different influences on enterprise sustainable innovation, and whether there are any interactions among the three types of searches requires clarification through further research. Second, this study only discussed the impact of boundary-spanning search on enterprise sustainable innovation from the perspective of enterprise capability reconfiguration. However, there may be other mediating variables, such as network location, system catch-up ability, and strategic agility, affecting enterprise sustainable innovation. Therefore, future research can comprehensively consider these variables to improve the theoretical research towards enterprise sustainable innovation. Third, this study mainly uses the panel data of manufacturing enterprises. However, enterprise sustainable innovation is a dynamic process, and the cross-sectional data collected through the questionnaire may have insufficient explanatory power. Therefore, researchers can track typical cases vertically in the future or expand the data collection batches and introduce time series analysis to make the research conclusions more universal and instructive.

**Author Contributions:** Conceptualization, N.C. and J.W.; methodology, N.C. and J.W.; software, L.Y.; validation, N.C., J.W., and Y.W.; formal analysis, J.W. and Y.W.; investigation, L.Y.; resources, N.C.; data curation, Y.W.; writing—original draft, N.C. and J.W.; writing—review and editing, Y.W.; visualization, L.Y.; supervision, N.C.; project administration, J.W. All authors have read and agreed to the published version of the manuscript.

**Funding:** This research was funded by the National Natural Science Foundation of China (grant number 71972050) and Key Support Program of Anhui Provincial University Excellent Talent (No.gxyqZD2019078).

**Institutional Review Board Statement:** Not applicable.

**Informed Consent Statement:** Not applicable.

**Data Availability Statement:** All data generated or analyzed during this study are included in the published article.

**Conflicts of Interest:** The authors declare no conflict of interest.

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
