# Peer review of "Towards Enterprise Sustainable Innovation Process: Through Boundary-Spanning Search and Capability Reconfiguration"

_processes, doi:10.3390/pr9112092_

Round 1
Reviewer 1 Report
Thank you very kindly for your paper. While there are many definitions of your major constructs, you have not explicitly stated your research question and the aim of the study in the introductory section. One would expect that you also discuss the key theoretical contributions in this section even though you have a separate sub-section for that. You need to focus your theoretical contributions on the novel ones your study is adding to the extant body of work. Any empirical contribution? Please, highlight this in the introductory section.
The hypotheses make sense, but you need to further connect your theoretical framework to them. It is not enough that you have a literature review. There should be a connection to your study.
Enterprise sustainable innovation is a very dynamic process, particularly when firms from emerging economies are examined. As such, cross-sectional data may not suffice. You should further discuss (justify)this in your methods section to convince the readers.
There's a need for some editorial work as well. Section five should be edited.
Reviewer 2 Report
The aim of the study is to explore the effect of boundary-spanning search on sustainable innovation in the manufacturing companies.
The purpose of the study is clear but needs to be better defined in the abstract. The results support interesting hypotheses for knowledge development. The conclusions of the article are well described and supported by the results of the analysis.
The introduction section and theoretical background are well-organized.
The methodology section should be implemented with some adjustments. Before presenting the subsections it would be appropriate to introduce a few lines on the method used for the analysis of the collected data by referring to the scientific literature on methodology.
For the subsection on sample and data collection (3.1) introduce a graph or table that summarises the information and makes it clear how the collection is done (with the number of responses related to each box).
The subsection on analysis (3.3.) needs justification in the literature for the threshold values taken as reference.
Justify the choice of the manufacturing sector and better highlight the gap in the literature.
The paper deals with an interesting and topical issue. The article is written in understandable English with concise and clear sentences.
